# ST09, A Novel Curcumin Derivative, Blocks Cell Migration by Inhibiting Matrix Metalloproteases in Breast Cancer Cells and Inhibits Tumor Progression in EAC Mouse Tumor Models

**DOI:** 10.3390/molecules25194499

**Published:** 2020-09-30

**Authors:** Snehal Nirgude, Raghunandan Mahadeva, Jinsha Koroth, Sujeet Kumar, Kothanahally S. Sharath Kumar, Vidya Gopalakrishnan, Subhas S Karki, Bibha Choudhary

**Affiliations:** 1Institute of Bioinformatics and Applied Biotechnology, Electronic city phase 1, Bangalore 560100, India; snehalnirgude3490@gmail.com (S.N.); raghunandan.hunsur@gmail.com (R.M.); kjinsha@gmail.com (J.K.); 2Manipal Academy of Higher Education, Manipal 576104, India; 3Department of Pharmaceutical Chemistry, KLE College of Pharmacy, Rajajinagar, Bangalore, (A Constituent Unit of KLE Academy of Higher Education & Research, Belagavi), Bangalore, KN 5600, India; klempharma@gmail.com (S.K.); subhasskarki@gmail.com (S.S.K.); 4Department of Studies in Chemistry, University of Mysore, Manasagangotri, Mysuru 570006, India; sharu.shivaramu@gmail.com; 5St. Joseph’s College, Irinjalakuda, Kerala 680661, India; vidyabc.iisc@gmail.com

**Keywords:** MDA-MB-231, MCF7, EAC mouse tumor model, apoptosis, migrastatic, immunity boosting cancer therapeutics

## Abstract

Purpose: Curcumin is known for its anticancer and migrastatic activity in various cancers, including breast cancer. Newer curcumin derivatives are being explored to overcome limitations of curcumin like low bioavailability, stability, and side effects due to its higher dose. In this study, the synthesis of ST09, a novel curcumin derivative, and its antiproliferative, cytotoxic, and migrastatic properties have been explored both in vitro and in vivo. Methods: After ST09 synthesis, anticancer activity was studied by performing standard cytotoxicity assays namely, lactate dehydrogenase (LDH) release assay, 3-(4, 5-dimethylthiazol-2-yl)-2–5-diphenyletrazolium bromide (MTT), and trypan blue exclusion assay. Annexin-FITC, cell cycle analysis using flow cytometry, and Western blotting were performed to elucidate cell death mechanisms. The effect on the inhibition of cell migration was studied by transwell migration assay. An EAC (Ehrlich Ascites carcinoma) induced mouse tumor model was used to study the effect of ST09 on tumor regression. Drug toxicity was measured using aspartate aminotransferase (AST), alanine aminotransferase (ALT), blood urea nitrogen (BUN), and flow-cytometry based lymphocyte count. Histological analysis was performed for assessment of any tissue injury post ST09 treatment. Results: ST09 shows an approximate 100-fold higher potency than curcumin, its parent compound, on breast tumor cell lines MCF-7 and MDA-MB231. ST09 arrests the cell cycle in a cell type-specific manner and induces an intrinsic apoptotic pathway both in vitro and in vivo. ST09 inhibits migration by downregulating matrix metalloprotease 1,2 (MMP1,2) and Vimentin. In vivo, ST09 administration led to decreased tumor volume in a mouse allograft model by boosting immunity with no significant drug toxicity. Conclusion: ST09 exhibits antiproliferative and cytotoxic activity at nanomolar concentrations. It induces cell death by activation of the intrinsic pathway of apoptosis both in vitro and in vivo. It also inhibits migration and invasion. This study provides evidence that ST09 can potentially be developed as a novel antitumor drug candidate for highly metastatic and aggressive breast cancer.

## 1. Introduction

Breast cancer is a heterogeneous and multifactorial disease. Breast cancer has been categorized based on its distinct morphological, histopathological, clinical, immunological, molecular, and genetic features affecting approximately 12% of women worldwide [1,2,3]. Current treatments include surgery, radiation therapy, hormonal therapy, immunotherapy, and targeted therapy. Despite all the available therapy, mortality due to breast cancer metastasis is increasing. Radiation therapy and surgery alone are of no great help, whereas chemotherapeutic options have significant side effects on healthy cells. Therefore, the search for more efficient, potent, and less toxic cancer treatment strategies is still in demand in cancer research.

Breast cancer therapy depends on the molecular classification leading to better survival outcomes [4]. This molecular classification thus provides an option to choose the right cell lines as a model and help to improvise on the available treatments along with the development of new drugs [5]. MCF7, T47D, and MDA-MB-231 are the most widely used cell lines in breast cancer research. All three cell lines are derived from pleural effusion but represent different molecular subtypes of breast cancer. MCF7 and T47D represent luminal A (ER+ve, PR+ve, and HER2-ve) type, whereas MDA-MB-231 represents the claudin-low subtype (ER-ve, PR-ve, and HER2-ve) [1,6]. MCF7 constitutively expresses receptors for estrogen, androgen, progesterone, and glucocorticoid, expanding its application for studying the regulation by all these hormones. T47D is ideal for the progesterone-specific response, whereas, in the presence of estrogen, MCF7 does not respond to progesterone [7]. MDA-MB-231 cells exhibit mesenchymal characteristics, express Vimentin, and are highly metastatic compared to non-metastatic MCF7, T47D cells [8].

Curcumin, a polyphenolic extract of rhizome *Curcuma longa*, is very well known for its anti-cancer properties in various cancers, including breast cancer [9,10]. It is imperative to point out that no toxicity was observed in either experimental animals or humans related to curcumin [11]. However, to overcome the limitations of low bioavailability and rapid metabolism, curcumin derivatives are being explored [6,12]. The importance of dimers of DAPs has been documented [13]. In this study, we have synthesized 1,2-bis((3E,5E)-3,5-bis(4-chlorobenzylidene)-4-oxopiperidin-1-yl)ethane-1,2-dione hemihydrate (C_40_H_28_Cl_4_N_2_O_4_½H_2_O, molecular weight 751.47), referred as ST09, whose anticancer potential has been explored.

ST09 is a molecular hybrid from the dimer 1,2-bis((3E,5E)-3,5-dibenzylidene-4-oxo-1-piperidyl)ethane-1,2-dione and is a structural isomer of the earlier reported derivative ST03 and functional derivative of ST08 [13]. Both ST03 and ST08 were potent derivatives of curcumin that exhibited cytotoxicity in the nanomolar range and induced cell death by activating the intrinsic apoptotic pathway. ST08 inhibits migration by downregulating MMP1. ST09, shown in Figure 1, is also one potent derivative of curcumin, which inhibits cell migration and is cytotoxic in the nanomolar range.

This study aims to understand the mechanism of ST09 induced cytotoxicity, cell death pathways, and inhibition of cell migration in vitro and toxicity and tumor regression in vivo. With minimal drug dosage and toxicity, we postulate that ST09 is a novel potential candidate for cancer therapy.

## 2. Materials and Methods

### 2.1. Chemistry

Thin-layer chromatography was performed with silica gel plates using toluene and ethyl acetate in a 1:1 proportion. The IR spectra were recorded in KBr using Jasco 430+ (Jasco, Tokyo, Japan); the 1H NMR spectra were recorded in CDCl3/DMSO using the Bruker(400 MHz NMR Spectrometer, Billerica, MA, US), and J values were reported in hertz (Hz). Mass spectra were recorded in triple quadrupole LCMS-6410 (Agilent technologies, Santa Clara, CA, United States).

### 2.2. Procedure for the Synthesis of ST09

Step 1: To the suspension of 4-piperidone hydrochloride monohydrate (13.03 mmol) in acetic acid (35 mL), 4-chlorobenzaldehyde (26.71 mmol) was added dropwise. To obtain a clear solution, dry hydrogen chloride gas was passed through this mixture. The reaction mixture was stirred at room temperature for 24 h and the precipitate was separated through filtration. The precipitate was added to a mixture of a saturated aqueous potassium carbonate solution (25% *w*/*v*, 25 mL) and acetone (25 mL) and was stirred for 0.5 h. The base obtained was washed with water (50 mL), and dried. To obtain a pure compound as hemihydrate, recrystallization was done using 95% ethanol.

Step 2: Dropwise addition of Oxaloyl chloride (0.003 mol, 0.39 g) in 1,2 dichloroethane (DCE) (5 mL) to a stirred suspension of a 3,5-bis(4-chlorobenzylidene)piperidin-4-one (0.006 mol) in DCE (20 mL) containing triethylamine (0.006 mol, 0.61 g) at 20 °C for a period of 30 min was carried out. After stirring at room temperature for 12 h, the solvent was removed under reduced pressure at 45 °C. Potassium carbonate (25 mL, 5% *w*/*v*) was added to the crude mass, stirred for 2 h, and the solid obtained was then filtered, dried, and crystallized from 95% ethanol to yield the pure product.

(ST09): Yield 50%, Rf 0.63, MP. 235–237 °C, IR (λ cm ^−1^) 3061, 2975, 2882, 1642, 1440, 1259, 1212, 1044, 990. ^1^H NMR (δ): 7.99 (s, 2H), 7.94 (s, 2H), 7.53–7.50 (m, 2H), 7.40–7.31 (m, 6H), 7.23–7.21 (m, 4H), 7.18–7.14 (m, 2H), 7.05–7.03 (m, 2H), 4.38 (s, 4H), 4.34 (s, 4H). MS (-ESI) *m*/*z*: 749.5 [M½H_2_O−2]

### 2.3. Chemicals and Reagents

Ponceau S (Sigma life Sciences), Phenylmethanesulfonyl chloride (PMSF), protease inhibitor cocktail tablets (EDTA-free), Tris base, Glycine, acrylamide, and bis-acrylamide powder and all other routine chemicals were purchased from MP biomedicals, (Santa Ana, CA, USA).

### 2.4. Cell Culture

MDA-MB-231, MCF7, and T47D cells were purchased from the National Centre of Cell Culture (NCCS), Pune, Maharashtra, India. MDA-MB-231 cells were grown in Dulbecco’s Modified Eagles Medium (DMEM high glucose with L-glutamine; Lonza), MCF7 in Eagle’s Minimum Essential Medium (EMEM; Lonza supplemented with non-essential amino acids (NEAA) from MP biomedicals), and T47D in Roswell Park Memorial Institute-1640 (RPMI; Lonza) media. All media were supplemented with heat-inactivated 10% fetal bovine serum (Gibco), 100 IU mg/mL penicillin/streptomycin (Gibco) at 37 °C in a humidified atmosphere containing 5% CO_2_. Peripheral blood mononuclear cells (PBMCs, a kind gift from SCR lab, IISc, Bangalore, India) and 293T cells were used as normal cells. ST09 was dissolved in DMSO such that all treatments had equal concentrations of dimethyl sulfoxide (DMSO) between 0.1–0.2%.

### 2.5. MTT Assay

The MTT assay was performed as described earlier [13]. Cells (5000 cells/well) were seeded in triplicates in 96-well plates. After a 24 h incubation, cells were treated with various concentrations of ST09. Cells were treated with MTT reagent, MP Biomedicals (5 mg/mL) at 37 °C, 5% CO_2_, after 24, 48, and 72 h of incubation with ST09 as described in [13]. Absorbance was recorded at 570 nm and the results shown were collected from three different biological replicates.

### 2.6. LDH Assay

The LDH release assay was done as described [13]. For this assay, 5000 cells were seeded in each well of the 96-well plate in triplicates. After 24 h incubation, cells were treated with varying concentrations of ST09. Curcumin, the parent molecule, was used for comparing the potency with ST09. After 24, 48, and 72 h of treatment, the absorbance of the orange-red colored formazan product was recorded at 490 nm and data from three different biological replicates were collected and plotted.

### 2.7. Trypan Blue Exclusion Assay

For the Trypan blue exclusion assay, 75,000 cells/mL cells were seeded in a 6-well culture plate as described earlier [13]. Treatment with varying concentrations of ST09 (1 to 1000 nM) was given after the 24 h incubation. After 48 h, collected cells were resuspended in 0.4% trypan blue (Sigma–Aldrich, St. Louis, MO, USA) and viable cells were counted using a hemocytometer. The percentage of viable cells was calculated and plotted as described earlier [13].

### 2.8. Cell Cycle Analysis

A total of 75,000 cells/mL of MCF7 and MDA-MB-231 were seeded in 6-well plates and treated with different doses of ST09 (0, 20, 40, 80, 100, and 120 nM) for 24 and 48 h. Cells were then trypsinized using TrpLE^TM^ Express Enzyme (1X, Gibco^TM^), washed twice by cold 1X PBS, and fixed in 80% cold ethanol overnight at −20 ° C as described [14]. Post fixation, cells were again washed twice with cold PBS followed by RNase A treatment (1 mg/mL RNase in 1X PBS) for 30 min at 37 °C. After staining with propidium iodide (PI, 1 µg/mL) for 30 min at 7 °C, cells were analyzed using a Gallios flow cytometer (Beckman Coulter, Miami, FL, USA). Data from three different biological replicates were collected and presented as bar graphs using GraphPad Prism 5 software. Cell cycle data were analyzed using Modfit LT free trial version 3.3 available from Verity software house. Cells were gated to include G0/G1, S-phase, and G2/M populations.

### 2.9. Apoptosis Assay

MCF7 and MDA-MB-231 cells (75,000/mL) were seeded in 6-well plates and were treated with different concentrations of ST09 (0, 20, 40, 80, 100, and 120 nM) for 48 h. Sample preparation for flow cytometry was performed as described earlier [13]. Ten thousand events were collected per sample using the Gallios flow cytometer (Beckman).

### 2.10. JC1 Staining

Mitochondrial potential change detection was done using a JC-1 (5,50,6,60-tetrachloro-1,10,3,30-tetraethylbenzimidazolcarbocyanine iodide) mitochondrial staining kit purchased from Sigma-Aldrich. MCF7 and MDA-MB-231 cells were treated with different concentrations of ST09 (0, 20, 40, 80, 100, and 120 nM) for 24 and 48 h. Cells were trypsinized and washed twice with cold PBS and the assay was done according to the manufacturer’s instructions. A minimum of 10,000 events were acquired using a Gallios flow cytometer (Beckman) for each sample, and data were represented as a dot plot.

### 2.11. Transwell Migration and Invasion Assay

A total of 7.5 × 10^4^ cells were seeded in a 6-well plate and were allowed to grow for 24 h followed by a 24 h treatment with 20 and 40 nM ST09. Permeable migration chambers were purchased from Corning Inc. (New York City, NY, USA; 24-well insert; pore size, 8 μm) and were coated with 75 µL of Matrigel (Corning) as described earlier [13]. Next, 5 × 10^4^ treated cells were suspended in 200 µL media without FBS and added into the top chamber. The cells were fixed with 4% paraformaldehyde and stained with 2% crystal violet after 5 h of migration. A cotton swab was used to clean the cells that did not migrate to the lower compartment. Every insert was imaged at 10× magnification, for five random fields and analyzed using NIH ImageJ software. Two independent experiments were carried out in duplicate.

### 2.12. Breast Adenocarcinoma Tumor Model

All the animal experiments were approved by the Institutional Animal Ethics Committee. EAC (1 × 10^6^ cells/animal) was injected to induce solid tumors in the left thigh region of Swiss albino female mice. After animals had developed a tumor of size ~200 mm^3^ the animals were segregated into 2 groups: control (*n* = 5) and ST09 treated (*n* = 5) as described in [15]. ST09 (10 mg/kg body weight (bd.wt)) was administered intraperitoneally (i.p) on alternate days for 12 days. The experiment was repeated at least three times with 5 animals in each group. Tumor size and body weight were measured for 25 days. Tumor volume was calculated using the formula V = 0.5 × a × b^2^, where V is tumor volume, and a,b are major and minor tumor diameters. Another group of animals (*n* = 5) was subjected to pre-treatment of 11 doses of ST09 (10 mg/kg of body weight) before inducing the tumor. After the 11th dose, tumors were induced as described above and the same procedure was repeated as the control and ST09 treated group. This group was designated as the “pre+post” treatment group. The study was approved by the “committee for the purpose of control and supervision of experiments on animals” (CPCSEA, Government of India, Animal welfare division, Reg.No. 1994/GO/ReBi/S/17/CPCSEA) and all experiments were performed following institutional, national guidelines and regulations of the CPCSEA.

### 2.13. Drug Toxicity and Side Effect Assessment on ST09 Treatment

EAC tumor-induced mice in the treatment group were treated with ST09 for 25 days and then were evaluated for drug toxicity. The pre+post treated animals were also evaluated for drug toxicity. Blood samples and serum were collected from animals from all treatments. The toxicity was assayed using standard enzymatic assays like AST, ALT, and BUN (Autospan, Span Diagnostics, Bengaluru, India) using the manufacturer’s prescribed methodology [15]. For checking changes in the hematological parameters, RBCs and WBCs were counted.

### 2.14. Histological Analysis of Tumor Tissues

Formalin-fixed tissues (tumor, liver, kidney, and spleen) from control and ST09 treated mice were processed as described earlier [15]. Sections were stained using Hematoxylin-Eosin (HE) and visualized at 10× magnification using a light microscope.

### 2.15. Immunoblotting

A total of 75,000 cells/mL were seeded and treated with ST09 (20, 40, 60, and 80 nM) for 48 h and the whole cell lysate was prepared as described [13]. Next, 30 µg of cell lysates was electrophoresed on 10 to 12% of SDS-PAGE (poly acrylamide gel electrophoresis) and were transferred to a polyvinylidene fluoride membrane (Millipore, Burlington, MA, United States). Blocking was performed using 5% skim milk in 1× PBS and then probed with primary antibodies: MMP2 from Biolegend, MMP1 from elabscience, Apaf, Bad, Bcl2, cytochrome c, Tubulin from Santa-Cruz Biotechnology, CA, and Caspase 9, Caspase 3, PARP, Vimentin, Bax, and GAPDH from Cell Signaling Technology, Beverly, MA, USA, followed by HRP-conjugated secondary anti-rabbit, anti-mouse antibodies (Cell Signaling Technology). The blots were developed using chemiluminescence reagent (Clarity Western ECL blotting substrate, Biorad) and the blot images were captured by the Chemidoc-XRS Biorad gel doc system. The protein band images were quantified using GelQuant.Net, BiochemLab solutions.

### 2.16. Flow Cytometry for Lymphocyte Analysis

The bone marrow of normal (*n* = 2) and 24 h ST09 treated animals (*n* = 2), were dissected and flushed with PBS to collect the cells. These cells were then subjected to flow cytometry analysis. A forward scatter (FSC) vs. side scatter (SSC) plot was used to analyze lymphocytes on the Gallios flow cytometer (Beckman Coulter, Miami, FL)**.** A minimum 10,000 events were acquired for each sample, and data were represented as a dot plot.

### 2.17. Statistical Analysis

For statistical analyses, GraphPad Prism 5 software package (San Diego, CA, USA) was used. To compare control versus treatment groups, Student’s t-test, two-way ANOVA, and Tukey’s test were performed. Values with a *p*-value less than 0.05 are considered as significant and statistical significance is represented as **** (*p*-value < 0.0001), *** (*p*-value < 0.001), ** (*p*-value < 0.01), * (*p*-value < 0.05, ns = not significant).

## 3. Results

### 3.1. Characterization of ST09

ST09 was prepared as per the procedure mentioned, and the structure of the synthesized compound (Figure 1A) was confirmed by IR, NMR, and mass spectrometry. In IR, the vibration of aromatic C–H bonds was observed between 3075 and 3003 cm^−1^, whereas for aliphatic C–H bonds were observed between 2946 and 2840 cm^−1^, and for carbonyl, C=O bonds were observed between 1666 and 1640 cm^−1^. In ^1^H NMR, ST09 showed prominent signals between δ 7.99 and 6.64 ppm for aromatic and olefinic protons. The structure of ST09 was also ensured by mass spectrometry. The mass spectrometry and ^1^H NMR characterization are shown (Figure 1B,C).

### 3.2. ST09 Exerts a Cytotoxic Effect on All Three Cell Lines with IC50 Values in the Nanomolar Range and Least Cytotoxicity on Normal Cells

To assess cell proliferation and cell viability, LDH, MTT, and trypan blue exclusion assays were performed on three breast cancer cell lines, 293T (normal kidney cell line), and PBMCs (peripheral blood mononuclear cells). ST09 exhibited cytotoxicity in the nanomolar range on breast cancer cells after 24, 48, and 72 h of treatment (Figure 2B,C), and IC50 values were calculated at the 48-h time point using both the assays and are tabulated in Table 1. Among the three cancer cell lines, the triple-negative breast cancer (TNBC) cell line MDA-MB-231 cells showed maximum cytotoxicity (IC50 0.031 µM). In contrast, luminal cancer cell lines MCF7 and T47D exhibited an IC50 of 0.093 and 0.138 µM, respectively, indicating that ST09, like curcumin [16], was more sensitive towards aggressive mesenchymal MDA-MB-231 cells. ST09 exhibited cytotoxicity at nanomolar concentration, 100-fold less than the parent compound curcumin [17] (Appendix A). Whereas minimal cytotoxicity was observed for both normal cells, PBMCs, and 293T, at 150 and 1000 nM of ST09 treatment, respectively (Figure 2A–C).

### 3.3. ST09 Arrests the Progression of Cell Cycle by G2/M Arrest in MCF7 and an Increase in the Sub-G1 Population in MDA-MB-231 Cells

The mechanism of ST09 antiproliferative activity and cell cycle changes were analyzed by performing flow cytometry after 24 and 48 h of ST09 treatment. ST09 exhibited both time and dose-dependent effects on cell cycle progression. As shown in Figure 3A, B, in MCF7 cells, the peak area representing the G2/M population increased to 57.08% after 48 h ST09 treatment compared to 12.48% in untreated control cells. MDA-MB-231 cells showed a different trend of cell cycle disruption on ST09 exposure. The area of the peak representing the hypodiploid (apoptotic, subG1/G0) population increased to 48.43% compared to 4.51% in untreated control cells (Figure 3C,D). These results show that ST09 induced a distinct effect on the cell cycle in a cell type-specific manner.

### 3.4. ST09 Induces Apoptosis in Human Breast Cancer Cells

The mode of cell death (apoptosis/necrosis) after ST09 treatment was analyzed using annexin/FITC-PI staining followed by flow cytometry. The upper and lower right quadrants indicate late and early apoptotic cells, respectively. It was seen that ST09 induced apoptosis in >50% MCF7 cells at 150 nM (Figure 4A,C) and MDA-MB-231 cells at 60 nM (Figure 4B,D) of ST09 in a dose-dependent manner. Early apoptotic cells were seen in MCF7 cells and late apoptotic cells in MDA-MB-231. No significant necrosis was seen in any of the cells post ST09 treatment, indicating that ST09 induced cell death via apoptosis.

### 3.5. ST09 Induced Change in the Mitochondrial Membrane Potential of Human Breast Cancer Cells

ST09 induced change in the integrity of mitochondrial membrane potential (MMP) was assayed using JC-1. It was observed that treatment with ST09 on MCF7 and MDA-MB-231 cells led to an increase in JC-1 green fluorescence in a dose and time-dependent manner. In MCF7 cells, the cell population exhibiting JC-1 green fluorescence increased gradually from 9.13% at 80 nM to 22.48% at 120 nM after 24 h ST09 treatment and from 25.01% at 80 nM to 63.32% 120 nM after 48 h ST09 treatment (Figure 5A,B). Similarly, for MDA-MB-231 cells, after 24 h ST09 treatment there was a gradual increase from 12.2% at 40 nM to 34.3% at 80 nM and from 32.95% at 40 nM to 72.6% at 80 nM after 48 h of ST09 treatment (Figure 5C,D). From the above experiments, it can be inferred that ST09 induces cell death by altering MMP and inducing the intrinsic apoptotic pathway.

### 3.6. ST09 Induced Intrinsic Pathway of Apoptosis in Human Breast Cancer Cells

After observing the apoptotic population and increase in JC-1 green population, immunoblotting was performed to understand the molecular mechanism of apoptosis following ST09 treatment on MCF7 and MDA-MB-231 cells. Activation of apoptotic markers by intrinsic pathways such as cleaved caspase 9, cleaved caspase 3, and cleaved PARP1 was observed (Figure 6A, C). The components of the apoptosome (Apaf + cytochrome C) were upregulated in both the cells on ST09 treatment (Figure 6A,C). Interestingly, Bax and Bad were upregulated, and Bcl2 was downregulated in MDA-MB-231 cells (Figure 6A,B). ST09 induced the mitochondrial pathway of apoptosis by activating pro-apoptotic proteins and inhibiting antiapoptotic proteins.

### 3.7. ST09 Inhibited Migration as Assayed by the Transwell Migration Assay

ST09 impact on cell migration was assayed using the Matrigel cell migration assay. MDA-MB-231 cells that exhibit mesenchymal cell morphology [18] treated with 20 and 40 nM ST09 for 24 h and untreated were added to migration chambers coated with Matrigel and allowed to migrate towards the chemoattractant. The migration ability was reduced ~five-fold in treated cells compared to untreated control cells (Figure 7A,B). To dissect the molecular mechanism of inhibition of cell migration by ST09, we examined MMP1, MMP2, and Vimentin expression levels. It is known that MMP1 [13,19], MMP2 [20,21], and Vimentin [22,23] have a significant role to play in migration and invasion. It was seen that ST09 significantly inhibited MMP1, MMP2, and Vimentin expression in MDA-MB-231 metastatic cells (Figure 7C). ST09 inhibited matrix metalloproteases and decreased Vimentin levels to block the migration of the cells.

### 3.8. ST09 Induces Tumor Regression in an EAC Induced Allograft Tumor Mouse Model

The tumor was developed in the left thigh region of Swiss albino mice by injecting EAC cells. Once all the mice had developed a tumor of ~200 mm^3^, they were segregated into two groups and one group of pre+post-treated animals in such a way that all the groups had animals with a similar range of body weight and tumor load. Thirteen doses of ST09 (10 mg/kg body weight) were administered every alternate day for 25 days, and the rate of tumor growth was monitored. Both ST09 pre+post and post-treatment resulted in significant tumor volume reduction compared to the control group (Figure 8A). ST09 pre+post treatment had the most significant tumor volume reduction. An anatomically similar trend was observed for tumors in both control and ST09 pre+post and post-treated groups (Figure 8A,B). An increase in the spleen size was observed in tumor control animals compared to ST09 pre+post and post-treated animals. The toxicity induced by ST09 treatment was examined by recording the body weight throughout the experiment time frame. The weights of the experimental animals at the end of the experiment period were plotted as a bar graph (Figure 8C) to investigate toxicity related weight loss, if any. There was no notable weight reduction due to the treatments in animals during the period.

### 3.9. ST09 Did Not Induce Observable Histopathological Changes

The liver, spleen, and kidney sections were checked for any visible changes in the morphology due to ST09 treatment. The tumor tissue and organs were dissected out at the end of the experimental period (25th day) and examined by H&E staining. The H&E stained tumor tissues demonstrated densely stained and packed nuclei in the untreated control tissue section, which shows a high amount of proliferating cells compared to the treated section. ST09 treated tumor tissues showed a reduction in proliferating cells along with fragmented nuclei indicating apoptosis. Similarly, kidney and spleen tissues were also checked for changes in cellular morphology by H&E staining (Figure 8D). No significant change in the morphology and integrity of cells was observed in any of the tissues. Hence our study shows that ST09 did not exert any significant adverse effect on organs and tissues.

### 3.10. ST09 Does Not Induce Any Toxicity and Does Not Have Any Side Effects

The effect of ST09 treatment on physiological parameters was tested by collecting blood samples of the tumor-bearing control, pre+post, and post-treated mice at the end of the experiment. We performed hematological, liver, and kidney functional assays to study the side effects on them. Serum was collected from all three groups of animals after 25 days of ST09 treatment and assessed for ALT, AST, and BUN. As shown in Figure 8E, the serum AST, ALT, and BUN levels were within the normal range in both the treatment groups (AST < 100 U/L and ALT < 60 U/L). No change in RBC count was observed between control and ST09 post-treatment (Figure 8F), but a slight decrease was observed in pre-treated animals. Interestingly, we noticed an increase in the number of WBCs in the ST09 post-treated groups compared to the control group (Figure 8F), which shows the immune activation property of the ST09 compound. Flow cytometry analysis of bone marrow showed a rise in lymphocytes for ST09 treated animals (Figure 8G).

### 3.11. ST09 Induces an Intrinsic Apoptotic Pathway In Vivo

The mechanism of tumor reduction in experimental animals was investigated by Western blot and immunohistochemistry. PCNA is a marker of proliferating cells. IHC with PCNA on control and treated tumors showed a large number of proliferating cells in control tumors compared to treated tumors (Figure 9A). Immunoblotting was done on protein extracted from ST09 and control tumor tissues. We observed a significant upregulation of pro-apoptotic proteins in the treated samples (Figure 9B,C). Proteins such as Bax, Apaf-1, and procaspase 9 were found to be upregulated. Upregulated PARP and its cleaved product were also detected in ST09 treated tumor tissue samples. A decrease in BCL2 expression was also observed. These results suggest that ST09 treatment led to a tilt in the balance of pro-apoptotic proteins to apoptotic proteins, which led to a reduction in tumor growth.

## 4. Discussion

In this study, the antiproliferative, migrastatic, and cytotoxic effect of ST09 in breast cancer cells and allograft mouse tumor models was investigated. ST09 treatment decreased the cell viability of breast cancer cells in both a dose and time-dependent manner while exhibiting the least toxicity towards normal cells. ST09 induced cell cycle arrest in epithelial MCF7 cells, whereas it induced sub-G1 hypodiploid population accumulation in TNBC MDA-MB-231 cells. Also, it activated an intrinsic apoptotic pathway both in vitro and in vivo. Interestingly, ST09 inhibited migration and invasion in most aggressive MDA-MB-231 cells by downregulating classical EMT markers like Vimentin, MMP1, and MMP2. Additionally, ST09 reduced tumor burden with no apparent toxicity and histological injury. Besides, a possible boost in immune response was observed in ST09 treated animals.

The development of breast cancer is a multistep process and is associated with disease relapse and drug resistance. Moreover, TNBC possesses a severe clinical challenge. TNBC does not have receptors for estrogen, progesterone, or human epidermal growth factor receptor 2 (HER2), which makes them unresponsive to endocrine treatment and other targeted therapies. TNBC has an aggressive clinical phenotype with a high metastatic rate and poor long-term prognosis [24]. Thus, the only viable treatment option that remains for TNBC is effective chemotherapy. Thus, novel drugs with minimal toxicity, higher potency, lesser side effects, and more bioavailability are needed to treat cancer. Plant-based therapeutics are thus being explored; phenolic extracts being one such class. Curcumin, proposed as an anticancer drug in 1985 [25], is one such example that has a broad spectrum of anticancer activity on several cancer types. Curcumin is also known to sensitize cancer cells to anticancer drugs [26,27]. However, due to several limitations like low water solubility, low cellular uptake, low bioavailability, low chemical stability, low absorption, and rapid metabolism; curcumin derivatives with structural modifications have been explored [28,29,30]. One such widely explored class is diarylidenylpiperidones (DAPs). DAPs share structural similarity with curcumin but have better bio-absorption with enhanced metabolic stability and bioavailability within the tissues compared to curcumin [30].

ST09, a structural isomer of ST03 and a functional isomer of ST08, was reported earlier [31]. However, they found that the response of the drug towards breast cancer cells was refractory. Also, the method of synthesis of the drug was different from what we have reported. In this study, ST09 has been synthesized from 2-bis ((3*E*,5*E*)-3,5- dibenzylidene-4-oxo-1- piperidyl)ethane -1,2-dione, by adding Cl-, a robust electron-withdrawing group, to the backbone. Previous studies have reported that the addition of electron-withdrawing groups in aromatic rings enhances the compound’s cytotoxic potential [32]. Like ST09, 3,5-Bis(2-chlorobenzylidene)-4-piperidone,3,5-Bis(4-chlorobenzylidene)-4-piperidone has enhanced antiproliferative potential, which can be attributed to ortho and para- substitution with Cl- [32].

ST09 induced a dose and time-dependent antiproliferative activity on three breast cancer cell lines with distinct molecular signatures [1]. ST09 exhibited differential cytotoxicity among two cell lines, just like curcumin [33], and showed increased sensitivity to MDA-MB-231 compared to MCF7, which suggests drug selectivity towards metastatic mesenchymal cells than luminal epithelial cells. A similar difference was observed in the cell cycle. MCF7 (epithelial cell) showed G2/M cell cycle arrest, whereas the sub-G1 population accumulation was observed for MDA-MB-231 (mesenchymal cells). Thus, ST09 induced cell cycle arrest leading to cell death in MCF7. In MDA-MB231, cell death was not associated with cell cycle arrest.

To understand the selectivity of ST09 towards cancer cells versus normal cells, PBMCs, and 293T cells were assayed along with breast cancer cells. Interestingly, ST09 is at least three times more cytotoxic towards cancer cells compared to normal cells (293T, PBMCs). ST09 showed the highest IC50 value of 138 nM towards T47D cells, whereas it was non-toxic at 150 nM on PBMCs and 1000 nM on 293T cells. Similarly, curcumin derivative (4c) showed the least toxicity in nonmalignant tissues, namely breast epithelial cells (MCF10A) and foreskin fibroblasts (Hs-27) [31]. This highlights the fact that ST09, like ST03 and ST08 [13], is selective and least cytotoxic on normal cells.

The modality by which the cell dies, whether necrosis or apoptosis, is critical for the outcome of cell death at the organismal level. Apoptosis, an immunogenic programmed cell death, eliminates unhealthy, unwanted cells and maintains surrounding tissue [34,35]. In cancer, evasion of cell death is one of the major mechanisms for abnormal cells to transform themselves into a malignant one [36]. Thus, drugs that can induce apoptosis, specifically in malignant and metastatic cells, would be the best cancer treatment candidates. Curcumin derivatives like EF24 and RL71 at a concentration of >2 µM and ST08 at a concentration of ~50 nM are known to induce an apoptotic response in human MDA-MB-231 cells [13,37,38]. ST09 exhibited significant antiproliferative activity and cytotoxicity towards MDA-MB-231 metastatic cells. ST09 induces apoptosis rather than necrosis at a ten-fold lower concentration than EF24, a potent curcumin derivative. ST09 mediated apoptosis by altering the mitochondrial membrane potential of breast cancer cells. Upregulation of pro-apoptotic proteins like cleaved caspase 3, cleaved caspase 9, PARP1, Apaf1, Bax, Bad, cytochrome-c, and downregulation of antiapoptotic protein-like Bcl2 suggests the involvement of the intrinsic apoptotic pathway. ST09, like ST03 and ST08, thus displayed its mode of action like curcumin, independent of the receptor status of the cell [33].

EMT (epithelial to mesenchymal transition) is a complicated cellular program and has been categorized into three types: type one is involved in embryogenesis, gastrulation, and neural crest formation; type two is required for tissue regeneration and wound healing; type three is associated with malignancy, invasion, and metastasis [36,39]. Unbalanced EMT pathways are the primary cause of metastasis, which is observed mainly in triple-negative breast cancer and makes it aggressive. Therefore, there is a need to identify drugs that can target molecules in the EMT pathway. Thus, we further explored the potential of ST09 to inhibit cell migration by performing a transwell migration assay. An approximate five-fold reduction in migration was observed after ST09 treatment suggesting its strong potential to inhibit migration and invasion. The classical and key players of metastasis, like MMP1, MMP2, and Vimentin were significantly downregulated upon ST09 treatment. This characteristic of ST09 is similar to curcumin, the parent compound, and another derivative, ST08, reported from our lab [13]. Curcumin regulates various key molecules like fibronectin, E-cadherin, Vimentin, Slug, Axl, Twist1, MMP1, and MMP2 in EMT in MDA-MB-231 cells and exhibits its migrastatic potential [40,41,42]. Thus, putting together ST09 at 100X lower concentration and better bioavailability is a potent drug candidate for metastatic, aggressive cancer therapeutics.

ST09 dosages of 10 mg/kg (13 doses) were sufficient to inhibit tumor growth in the tumor allograft tested. The tumor regression was significantly higher in pre+post treated animals than post-treatment alone. The enhanced effect might be due to the activation of immune response on drug treatment, as evidenced by an increased lymphocyte count [43]. Whereas curcumin is reported to exhibit its antitumor activity in vivo at 800 mg/kg of body weight, and when encapsulated with dendrosome, the effective dosage reduced to 40 mg/kg of body weight [44,45]. Another curcumin derivative, ST06 silver nanoparticles, reported from our lab, exhibited excellent potency at the lowest dosage of 5 mg/kg of body weight [15]. Put together, ST09 at lower dosage has a better impact on tumor reduction compared to curcumin. Also, lower concentrations of the drug might reduce off-target events [46]. ST09 induced minimal organ toxicity and side effects, partially due to the lower dosage. H&E staining showed densely stained and packed nuclei in the untreated control tissue section as compared to the treated tissue section, which indicates proliferating cells were mainly targeted by ST09. Interestingly, intrinsic apoptotic pathway markers like Apaf1, Bax, and procaspase 9 were upregulated in treated tumor samples compared to untreated control tumor tissue. Our results show that ST09 treatment induces cytotoxicity through the intrinsic pathway of apoptosis and reduces tumor growth in EAC injected tumor mouse models without adverse effects. Thus, ST09 can be considered and developed as a novel chemo-preventive and anticancer agent.

## 5. Conclusions

For the first time, we report the antiproliferative and cytotoxic mechanisms of action of a novel curcumin derivative ST09 in human breast cancer cells. ST09 is ~100× more potent than curcumin. It induces the intrinsic pathway of apoptosis and inhibits cell migration in vitro and in vivo. This study documents potency of ST09 against highly metastatic, recurrent, and invasive breast cancer and its potential to be developed as cancer therapeutics.

## Figures and Tables

**Figure 1 molecules-25-04499-f001:**
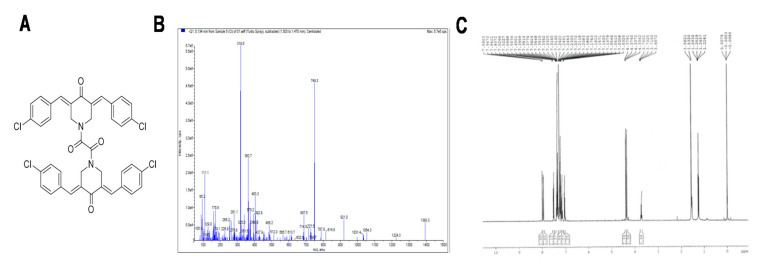
Characterization of ST09. (**A**) Structure of ST09; (**B**) mass spectrometry data of ST09; (**C**) 1HNMR data of ST09.

**Figure 2 molecules-25-04499-f002:**
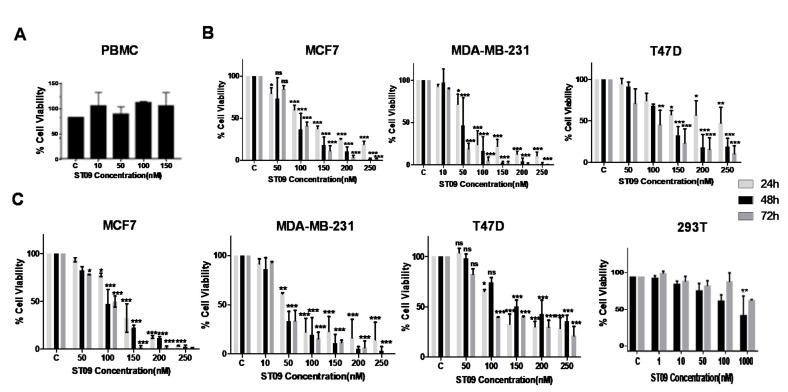
ST09 induced cytotoxicity analysis. (**A**) Viable PBMC cell count after ST09 treatment by trypan blue exclusion: the cells were collected and stained for trypan blue uptake after 48 h treatment with ST09. The unstained viable cells were counted and the percentage of viable cells in each dosage was calculated by considering vehicle control (DMSO) treated cells as 100% viable. (**B**) Evaluation of cell viability after ST09 treatment by LDH assay: bar graph depicting cell viability upon ST09 treatment on three cell lines as tested by LDH assay. (**C**) Evaluation of cell viability after ST09 treatment by MTT assay: bar graph depicting cell viability upon ST09 treatment on cell lines as tested by MTT assay. Data from three different biological replicates were collected for all experiments and were presented as bar graphs using the Graph Pad Prism tool. Two-way ANOVA followed by Tukey’s multiple comparison test was carried out and significance is represented as **** *p*-value < 0.0001, *** *p*-value < 0.001, ** *p*-value < 0.01, * *p*-value < 0.05. ns = nonsignificant.

**Figure 3 molecules-25-04499-f003:**
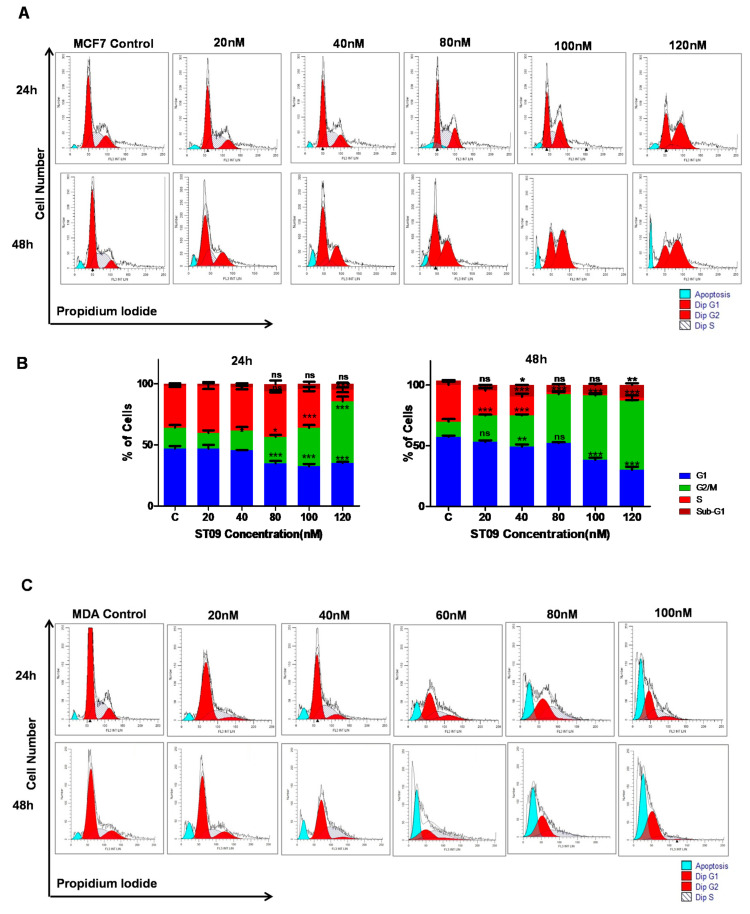
Impact of ST09 on the cell cycle in breast cancer cells: cell cycle profile of MCF7 (**A**), and MDA-MB-231 cells (**C**) at 24 and 48 h. Quantification of the percentage of cells in each phase of the cell cycle MCF7 (**B**) and MDA-MB-231 (**D**) cells is depicted as a bar graph of mean ± SEM at 24 and 48 h. Each experiment was done in triplicate and represented as histograms. A two-way ANOVA test was performed, and the *p*-value was calculated between control and ST09 treated groups (* *p* < 0.05, ** *p* < 0.005, *** *p* < 0.0001, ns = nonsignificant).

**Figure 4 molecules-25-04499-f004:**
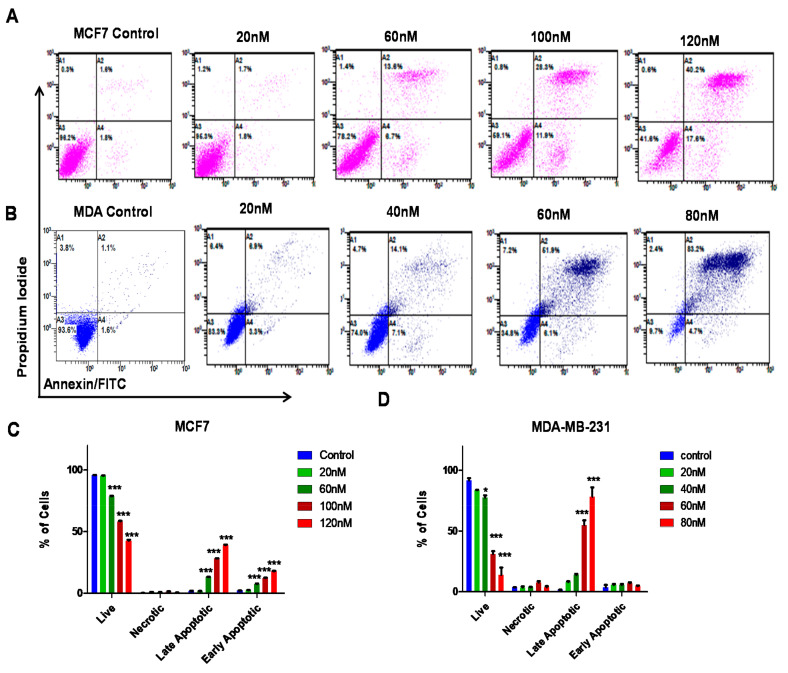
Evaluation of ST09 induced cytotoxicity on breast cancer cells by annexin-FITC/PI double staining: dot plot depicting MCF7 (**A**) and MDA-MB-231 (**B**) cells treated with ST09 (20, 40, 60, 80, 100, 120 nM) for 48 h. Quantification of MCF7 (**C**) and MDA-MB-231 (**D**) apoptotic cells. The experiment was repeated twice. A two-way ANOVA test was performed, and the *p*-value was calculated between the control and ST09 treated groups (* *p* < 0.05, *** *p* < 0.0001).

**Figure 5 molecules-25-04499-f005:**
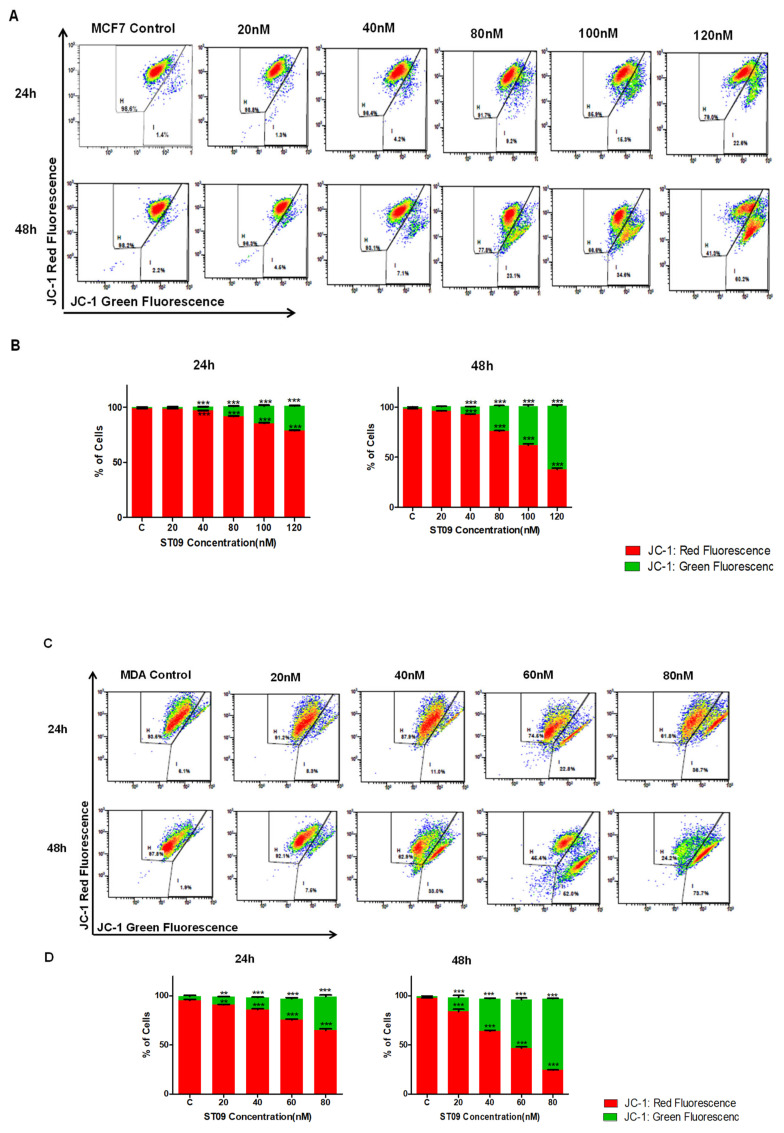
Evaluation of the mitochondrial membrane potential (MMP) changes induced by ST09 in breast cancer cells on JC-1 staining: dot plot depicting 24 and 48 h treated MCF7 (**A**) and MDA-MB-231 (**C**) cells by ST09 subjected to JC-1 staining. Quantification of high and low MMP is depicted as a bar graph after 24 and 48 h treatment of ST09 on MCF7 (**B**) and MDA-MB-231 (**D**) cells. Every experiment was repeated three times and represented as histograms. A two-way ANOVA test was performed, and the *p*-value was calculated between the control and ST09 treated groups (** *p* < 0.005, *** *p* < 0.0001).

**Figure 6 molecules-25-04499-f006:**
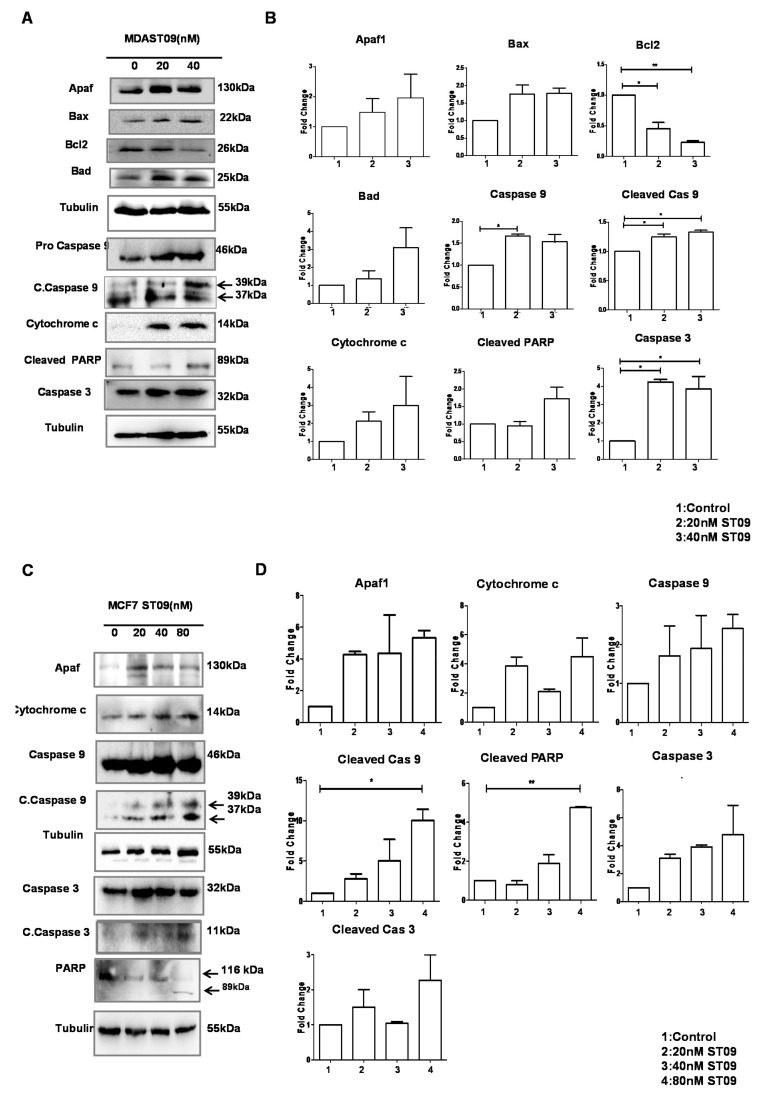
Assessment of apoptotic proteins in breast cancer cells treated with ST09. Western blot analysis of apoptotic markers in MDA-MB-231 (**A**,**B**) and MCF7 (**C**,**D**) cell lysates treated with different concentrations of ST09 (0, 20, 40, 80 nM) for 48 h. Each experiment was done in duplicate and a representative image is shown for each marker. Quantification was done for each marker and is represented as a bar graph of mean +/− SEM. A one-sample t-test and one-way ANOVA test were performed and the *p*-value was calculated between control and ST09 treated groups (*: *p*-value < 0.05, **: *p*-value < 0.005).

**Figure 7 molecules-25-04499-f007:**
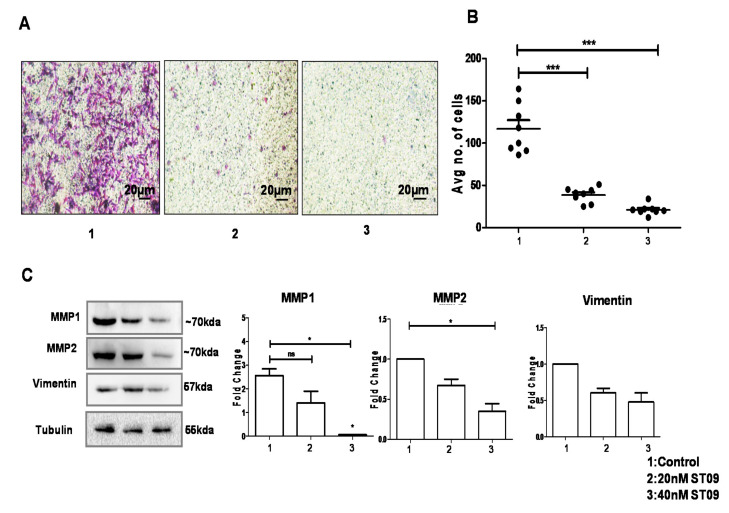
Effect of ST09 on cell migration: (**A**) ST09 treated MDA-MB-231 cells in vitro subjected to migration on Matrigel. (**B**) Quantification of migrated cells is represented as a scatter plot. (**C**) An immunoblot of MMP1, MMP2, and Vimentin proteins was performed from MDA-MB-231 cell lysate treated with the ST09 compound for 48 h. Quantification was done for each protein and is represented as a bar graph of mean +/− SEM. Every experiment was performed twice, and a two-way ANOVA test was used to calculate the significance between control and ST09 treated groups. Significance for immunoblots was calculated using a t-test and one-way ANOVA to calculate *p*-value between control and ST09 treated samples. (* *p* < 0.05, *** *p* < 0.0001).

**Figure 8 molecules-25-04499-f008:**
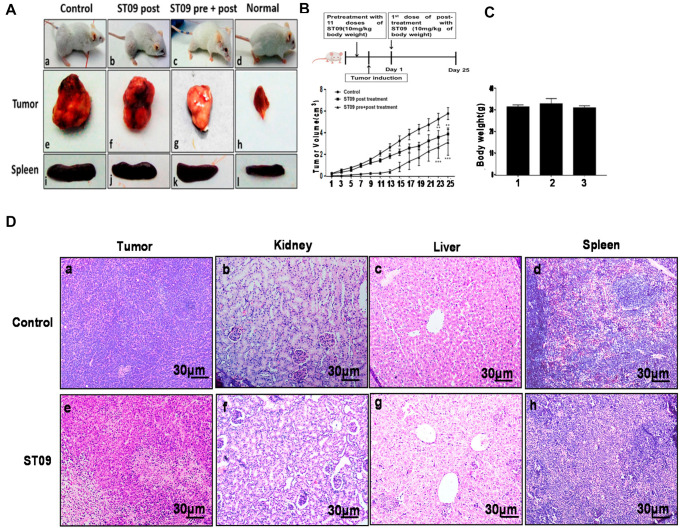
Evaluation of tumor reduction and organ toxicity in vivo induced by ST09: EAC cells (1 × 106 cells/ animal) were injected to induce solid tumors. After the 7th day of injection, i.p injection with ST09 (10 mg/kg b.wt) was started every alternate day throughout the experiment period. For the pre+post treatment experiment, animals were pre-treated with 11 doses of ST09 (10 mg/kg b.wt), the tumor was induced, and the treatment was continued. (**A**) The gross appearance of a. control, b. ST09 post-treatment, c. ST09 pre+post treatment, d. normal mice e. control tumor, f. ST09 post-treatment tumor, g. ST09 pre+post treatment tumor, h. normal thigh, i. Control spleen, j. ST09 post-treatment spleen, k. ST09 pre+post treatment spleen, l. normal spleen. (**B**) Tumor volume after ST09 pre+post and post-treatments. (**C**) Bodyweight of animals at the end of the study. (**D**) Histopathological analysis of tumor and organs after ST09 treatment. At the end of the study, tumor tissue and organs were collected and used for histological analysis. Representative images of H&E stained sections of a. control tumor, e. ST09 post-treatment tumor b. control kidney, f. post-treatment kidney, c. control liver, g. ST09 post-treatment liver d. control spleen, h. ST09 post-treatment spleen. (**E**) Blood ALT, AST, and urease test results were plotted as a bar graph. Blood was collected at the end of the study. (**F**) WBC and RBC counts of experimental animals plotted as a bar graph. (**G**) Flow cytometry analysis of bone marrow cells showing forward scatter vs. side scatter plot. Lymphocytes were gated and analyzed. Quantification of lymphocyte response is plotted as a bar graph post ST09 treatment. Each experiment was repeated twice and represented as histograms.

**Figure 9 molecules-25-04499-f009:**
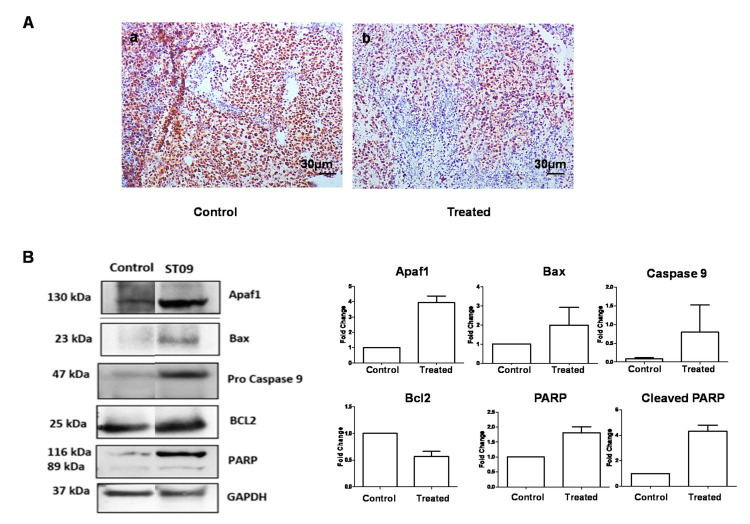
Immunohistochemistry and Western blot analysis post ST09 treatment in vivo. (**A**) PCNA stained control tumor(a) and treated tumor(b). (**B**,**C**) Western blot and quantification of pro and anti-apoptotic protein markers. A one-sample t-test and one-way ANOVA test was performed, and the *p*-value was calculated between control and ST09 treated groups.

**Table 1 molecules-25-04499-t001:** IC50 (µM) values of ST09 for three breast cancer cell lines are tabulated for 48 h treatment.

Cell Line	IC50 (µM)
MDA-MB-231	0.031
MCF7	0.093
T47D	0.138

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
