# Peer review of "ST09, A Novel Curcumin Derivative, Blocks Cell Migration by Inhibiting Matrix Metalloproteases in Breast Cancer Cells and Inhibits Tumor Progression in EAC Mouse Tumor Models"

_molecules, 2020, doi:10.3390/molecules25194499_

Round 1

Reviewer 1 Report

This study investigated the antiproliferative, cytotoxic and migrastatic properties of ST09, a novel curcumin derivative, in mouse tumor model and in human breast cell lines by using in vivo and vitro assay.  They concluded that ST09 has potential for further development as a novel antitumor drug candidate for highly metastatic and aggressive breast cancer.  

The topic of this study is very interesting and contribute to development of new anticancer agent for highly metastatic and aggressive breast cancer. In my opinion the paper can be accepted after a revision

Ihave itemized my major concerns in the following paragraphs.

General remarks:

  • It is necessary to carefully check all acronyms (i.e. EAC, the name of cell lines, MTT, etc) in the manuscript ( from abstract to conclusion) to make the text understandable to the reader. For molecule ST09 add the formula in the text also. Please specify the meaning of the acronym when it first appears in the text and then use acronym.
  • It is necessary enlarge all figures of this paper. In some figures, it is impossible to read the text in particular for the graphs and also for the westen blot. In general, all figures need to be enlarged and rearranged. the reader must be able to see and read.
  • delete the colon from the title of the various paragraphs throughout the manuscript

Specific remarks:

  • Lane 62-70: Improve this paragraph by adding and integrating with further information regarding curcumin and its derivative.

  • Lane 71-74:reinforce the aim of this research.

  • Lane 208: tatistical analysis paragraph. Improve this paragraph because Tukey’s test for multiple comparison is mentioned in the figure legends but there isn’t in this paragraph. Please add these informations.
  • fig.1. I think that it’s better to divide this figure in two figures. In particular panel A, B and C (enlarged) of this figure regarding the synthesis and structure of ST09 can became fig.1 and panel D, E and F (enlarged and rearranged) can became fig. 2 regarding cell viability. Enlarge the graphs and then for representation of the statistical p value , probably, it’s better to use the system with different letters on the bar graph instead of the asterisks. Ns = not significant???...there isn’t in the figure legend. If you divided this figure remember change the number in all figures .

  • Lane 248-250: It’s better to move this sentence in the discussion.
  • fig.2:Enlarge the graphs and then for representation of the statistical p value , probably, it’s better to use the system with different letters on the bar graph instead of the asterisks.
  • fig.3Enlarge the graphs (B, D ) and rearrange (there is an overlapping )
  • fig.4Enlarge all panels in this figure. It’s impossible to read in particular on hard copy
  • fig.5 :Enlarge all panels. Add molecular weight on the western blot image
  • Lane331: Add the number in the figure legend (6?)and enlarge all. In particular in the panel A add the magnification bar on the images and resize the images (same size for all)
  • fig. 7 :Enlarge and rearrange this figure because there are different problems. The images of all panels seem distorted. For panel D, add magnification bar and improve the quality of e.e. images.
  • Lane 391: delete the extended form of acronyms. Here you can use only acronims.

Author Response

Dear Reviewer,

We would like to thank you for the compliments and very thoughtful suggestions. We would like to apologize for the inconvenience caused due to small and unclear images. Also, these changes have been highlighted in the manuscript.

The detailed response is attached.

Regards,

Dr. Vibha Choudhary

Reviewer 2 Report

The study from Nirgudea et al investigates the anti-proliferative, migrastatic and cytotoxic effect of the curcumin derivative ST09 in breast cancer cells and in allograft mouse tumor models. They conclude that ST09 treatment decreased cell viability of breast cancer cells in both dose- and time-dependent manner while exhibiting less toxicity towards normal cells. Overall the study is innovative, well designed and clear. A few improvements are suggested to support data and help manuscript reading and interpretation.

- I was unable to read the figures. Their size is too low and graph labels are impossible to read.

- One figure comparing the effects of curcumin and ST09 (at least one experiment) would support the ST09 higher potency claimed in the paper.

- Figure 1 should be better divided into two, the first showing the chemical structure and analysis, and a second showing the compound toxic effect on cells.

- The reasoning of using 293T and PBMC cells for cytotoxic analysis is not well explained in the discussion. Figures too small, with poor resolution.

- Most of the experiments were done in duplicate. A higher number of tests is needed to perform statistical analysis. Statistical denotation p <= is not correct, change to the proper symbol.

- ST09 induced a distinct effect on the cell cycle in a cell type-specific manner. How can this distinct effect be explained, are there different mechanisms for each cell type?

Author Response

Dear Reviewer,

We would like to thank for the compliments and thoughtful suggestions given by you. We have rectified the manuscript as suggested. Also, these changes have been in the manuscript.

The detailed response has been attached.

Regards,

Dr. Vibha Choudhary

Round 2

Reviewer 2 Report

The authors replied to all the raised questions and improved the manuscript.